# A Method of Node Layout of a Complex Network Based on Community Compression

**Chengxiang Liu [1], Wei Xiong [2,*], Xitao Zhang [3] and Zheng Liu [1]**

[1] Department of Space Information, Space Engineering University, Beijing 101416, China; 201311141033@mail.bnu.edu.cn (C.L.); neverlinever@163.com (Z.L.)

[2] Science and Technology on Complex Electronic System Simulation Laboratory, Space Engineering University, Beijing 101416, China

[3] Department of Space Command, Space Engineering University, Beijing 101416, China; zxthl0707@163.com

[*] Correspondence: 13331094335@163.com; Tel.: +86-18518619059

**Abstract:** As the theory of complex networks is further studied, the scale of nodes in the network is increasing, which makes it difficult to find useful patterns from only the analysis of nodes. Therefore, this paper proposes a complex network node layout method based on community compression, which can effectively display the mesoscale structure characteristics of the network, making it more convenient for users to analyze the status and function of a single node or a class of nodes in the whole complex network. To begin with, the whole network is divided into communities with different granularity by the Louvain algorithm. Secondly, the method of nodes importance analysis based on topological potential theory is extended from the network to the community structure, and the internal nodes of the community are classified into three types, namely important nodes, relatively important nodes, and fringe nodes. Furthermore, a compression algorithm for the community structure is designed to realize the compression of the network by retaining important nodes and merging fringe nodes. Finally, the compression network is laid out by the traditional force-directed layout method. Experimental results show that, compared with the compression layout methods of a complex network based on degree or PageRank, the method in this paper can retain the integrated community composition and its internal structure, which is convenient for users to effectively analyze the topology structure of a complex network.

**Keywords:** complex network; community compression; node layout; node importance

## 1. Introduction

The purpose of network visualization is to assist users to perceive the network structure and understand and explore patterns hidden in the network data [1,2]. However, the details of all the nodes and edges, which are shown in their entirety on a limited-size screen, is not a good solution, and it may cause the following problems from a visualization point of view. On the one hand, when a complex network with large numbers of nodes and high-density edges is visualized on a limited-size screen, users fall easily into a chaotic and overlapping node-connection diagram, where it is difficult to identify and perceive the overall structure of the network, and the users cannot easily find elements of interest. On the other hand, the large amount of data inevitably brings greater computational pressure and higher requirements on the algorithms and hardware. In short, inefficient or time-consuming network visualization loses its meaning.

Until now, the existing work of complex networks has been limited to the local scale structure, obtained through statistical distribution, or the macroscopic scale of the overall parameters of the network. However, these two levels can be understood through the intermediate level, which is

called the mesoscale structure [3]. A mesoscale can be understood as a substructure or subgraph. Substructures have their own topological entities, such as motifs, cliques, cores, and loops, but usually they describe the community [4]. The research of network synchronization processes actually involves studying the evolution process of the dynamic behavior of the network from a small to a large scale, which is inseparable from the mesoscale characteristics of complex networks [5].

In order to display information effectively and assist users in perceiving the mesoscale structure of the network, a large-scale network must be reduced to bring down the user's perception complexity and the computational complexity of the layout process. According to the purpose of reduction, the methods of network reduction processing can be divided into three types: filtering, sampling, and compression. Filtering means to filter the nodes and edges that satisfy the condition by a single attribute or a combination of multiple attributes in the network, thereby reducing the number of nodes and the density of the edges. Filtering technology is more about further refinement and exploration of network data, which needs a certain understanding of the network structure. For example, the elements that meet the requirements are selected and displayed on the screen, according to the degree, the intermediate centrality, the intermediate centrality of nodes, the weight of edges, or any other attributes. Sampling is to select the representative nodes and edges according to a certain sampling strategy and to reflect the structural characteristics of the original network with as few nodes and edges as possible. Common sampling strategies [6] include random node selection, random edge selection, random walk, and sampling strategies based on topological layered models. Compared with filtering and sampling, compression is achieved by aggregating nodes or edges with certain similarities and adopting new nodes instead of aggregation [7]. While reducing the scale of the nodes, it can also ensure the integrity of the network and display the subgraphs of the network. The composition and other characteristics help users perceive the overall structure of the network from the mesoscale structure [8].

For the purpose of effectively displaying mesoscale structure characteristics of the network, a method of node layout of a complex network based on community compression is proposed, which combines the force-directed layout algorithm with the method of network compression based on node importance in community structure. The method proposed in this paper can be used to visually observe and analyze the mesoscale structural features of the network, understand the rules and patterns within the network from the mesoscale, and provide support for the research of the synchronization process of the community network.

## 2. Related Work

In the field of network visualization compression layout, the spring model [9] has been most widely explored because of the advantages of simplicity, efficiency, and easy understanding. It simulates the whole network as a spring system, and finally achieves the force balance of the nodes by calculating the force controlled by the distance between different nodes. The KK (Kamada-Kawai) algorithm [10] introduces Hooke's law on the basis of the spring model, calculates system energy according to the stress state of the nodes, and transforms the optimal layout problem of nodes into the solution problem of minimizing system energy, which makes the convergence speed of the layout process obviously increase. In the DH (Davidson-Harel) algorithm [11], the constraints of many aesthetic standards are taken into consideration to construct the energy function, such as node location, length of joints, and the intersection of edges. Different layout effects can be achieved by the parameters of the energy function model. In the FR (Fruchterman-Reingold) algorithm [12], the nodes are modeled as charged particles in the physical system, and the charge repulsion forces are introduced between particles. The smaller the distance between the particles, the greater the repulsion force is, which can reduce the overlap phenomenon of the dense nodes to a certain extent. The simulated annealing algorithm was adopted by introducing a "cooling function" in order to make the algorithm converge quickly, which can gradually reduce the moving step of the nodes and makes the system energy decrease rapidly, thus realizing the goal of rapid layout [13,14].

Compared with the KK and DH algorithms, the rendering results from the FR algorithm have better symmetry and clustering layout effects in most network visualization applications, and are more in line with aesthetic standards. Many scholars have further studied the basis of the FR model. On the one hand, it proposes a fast multi-scale layout [15], multi-level layout [16], and FADE [17] algorithm to reduce visual confusion and accelerate layout. On the other hand, it proposes a network visualization method based on community structure [18] or subgroup analysis [19]. The above method shows the community composition of the network to a certain extent, but it is far from sufficient to analyze the mesoscale features.

Studies on network compression are mainly based on the importance of nodes or edges. Saha et al. [20] studied the nearest neighbor search problem in complex networks by proposing an appropriate concept of proximity degree. Applying it to the problem of network community detection, it can explore the community structure of real networks better. This gave us the idea of studying the reasonable compression layout scheme of community structure. Wang et al. [21] proposed a sparse representation of network structure based on geometric multi-scale analysis, which effectively helps to analyze the network with as little information as possible. Li et al. [22] divided network nodes based on the K-core concept in complex networks, and used the force-directed layout algorithm to realize the visualization of a compressed large-scale complex network. However, the above methods evaluate the importance of nodes based on the global network, choose the important nodes as the representative nodes of the overall network structure, and compress the non-important nodes at the same time, so as to achieve the goal of network compression. This kind of method retains the core structure of the network from the node scale (microscale), but it ignores the community structure of the network and cannot show the structural characteristics of the network from the mesoscale.

Based on data fields, the authors of [23] described the potential values of nodes in the network affected by the nodes themselves and their neighboring nodes by calculating their topological potentials, which can more truly describe the importance of nodes in the network structure. The topological potential is applied to a variety of real network data in [24] and is compared with traditional node importance measures such as degreed, betweenness, and PageRank. It was pointed out that the topological potential can highlight the difference in node location in the network and reflect the importance of the node, while considering the importance of node degree.

Similarly, the contribution of different nodes in a community is different to the community structure. The greater the degree, and the closer the node is to the community center, the more important it is. Therefore, this paper considers the location of nodes in the community structure, and the importance of the evaluation of nodes based on topological potential is applied to the community structure according to the degree of nodes and the shortest path length between nodes. By choosing the nodes with high importance in the community structure as the representative nodes of the community structure, the compression of the network community structure is realized.

## 3. Method

### 3.1. Multi-Granularity Community Structure Detection

Compared with other community structure detection algorithms, the Louvain algorithm [25] has a faster operation speed and can quickly process a network with hundreds of millions of nodes [26,27]. In addition, hierarchical clustering can output different granularities of the community structure partition results. Therefore, this paper adopts the Louvain algorithm to realize multi-granularity community structure detection.

Modularity is one of the important indicators to describe the quality of community division in the network. It can be calculated by the following formula [25]:

$$Q = \frac{1}{2m}\sum_i\sum_j (A_{ij} - \frac{k_ik_j}{2m})\delta(c_{v_i}, c_{v_j}). \tag{1}$$

In the formula: $m$ is the total number of edges in the network; $A_{ij}$ is the weight of the edge between certain node pairs; $k_i$ is the degree of node $v_i$; $c_{v_i}$ is the community to which the node $v_i$ belongs; and $\delta(c_{v_i}, c_{v_j})$ is the Dirac function, which will be 1 if the nodes $v_i$ and $v_j$ belong to the same community and will otherwise be 0.

The community structure detection algorithm based on modularity optimization is a kind of aggregation algorithm, which continuously aggregates nodes by optimizing the modularity gain function, and finally obtains the result of community structure partition when the modularity gains stops increasing. In [25], the authors define the modularity gain as:

$$\Delta Q = \left[ \frac{C_{in} + K_{i,in}}{2m} - \left( \frac{C_{out} + K_i}{2m} \right)^2 \right] - \left[ \frac{C_{in}}{2m} - \left( \frac{C_{out}}{2m} \right)^2 - \left( \frac{K_i}{2m} \right)^2 \right]. \tag{2}$$

In the formula: $C_{in}$ is the sum of all internal edge weights in the community $C$; $C_{out}$ is the sum of all edge weights adjacent to the nodes in the community $C$; $K_i$ is the sum of edge weights of all adjacent nodes $v_i$; and $K_{i,in}$ is the sum of the weights of the edges to which all nodes ($v_i$) and the nodes in the community are adjacent.

As shown in Figure 1, the Louvain algorithm can be divided into two stages. Firstly, each node is initialized as one community, and all nodes are traversed continuously in the network. The generated modularity gains are calculated by adding the node to each community. If the modularity gain is greater than 0, the community with the largest gain of corresponding modules is selected from these corresponding communities and merged with them. The process is repeated until the two communities in the network are no longer merged. Then, based on the results of community division in the first stage, each community is abstracted as a "node" and a new network is constructed. The weights between new nodes are the sum of the weights between communities in the original network. The process of the previous stage is repeated until the communities can no longer be merged, that is to say, the modularity gain is no greater than 0.

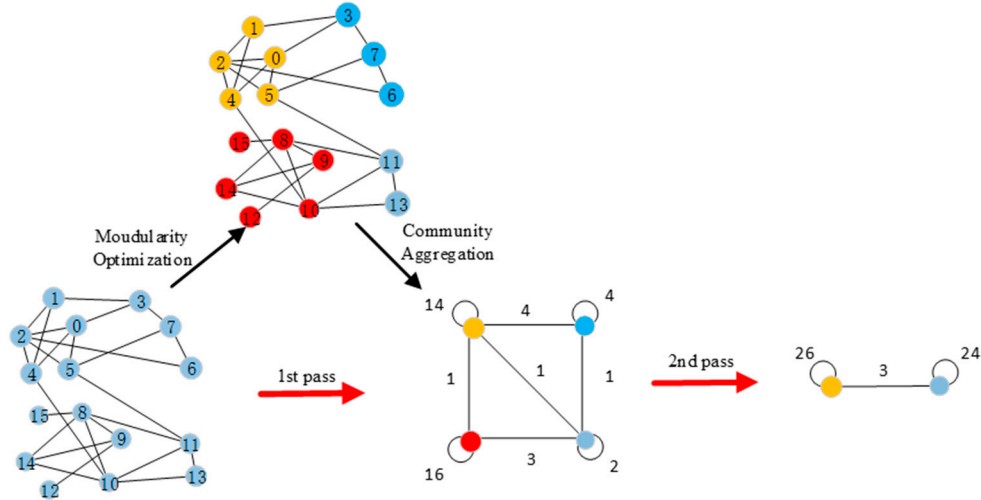

**Figure 1.** Schematic diagram of the Louvain algorithm.

## 3.2. Importance Assessment of Nodes in a Community Structure Based on Topological Potential

The value of node topological potential describes the potential of a node in the network topology affected by itself and its neighbors. Similarly, for a given community, $C = (V_C, E_C)$, the formula for calculating the topological potential of any node $v_i$ in the community is

$$\varphi(v_i) = \sum_{j \neq i}^{|V_C|} \left( k_j e^{-\left( \frac{d_{ij}}{\delta} \right)^2} \right). \tag{3}$$

In the formula: $|V_C|$ is the number of nodes in the community $C$; the larger $k_j$ is, the greater is the effect on the nodes; $d_{ij}$ is the shortest path length from node $v_i$ to node $v_j$; and $\delta$ is the influence factor, indicating the range of interactions between nodes.

Formula (3) shows that the potential of any node in a community is actually equal to the sum of the forces exerted on it by all other nodes in the community, and the forces gradually decay with an increase in the distance from the node. The more nodes around a certain node in the community and the shorter the distance is, the greater is the potential of the node. Therefore, it can be judged that the node with the highest potential value is the node at the center of the community, which can be regarded as the center of the community; conversely, the node at the fringe of the community usually has fewer edges and relatively low potential value.

According to the degree of the node and the shortest path length from the node to the center node of the community, the topological location of nodes in the community structure reflects the contribution of nodes to the community structure. As shown in Figure 2, the importance of nodes is assessed by calculating their potential in the community structure. According to the order of importance, the internal nodes of the community are divided into three categories: (1) Node 9 is the most important one as it has extreme topological potential and is the center node of the community, being marked with red on the left subgraph of Figure 2 and expressed with $V_{CA}$; (2) Nodes 7, 19, 21, 22, 25, 26, 10, and 16 have a great influence on the center node of the community, marked with yellow on the left subgraph of Figure 2 and expressed with $V_{CB}$; (3) The remaining nodes are fringe nodes, marked with gray on the left subgraph of Figure 2 and expressed with $V_{CC}$. The center node of the community is chosen as the representative point, and the nodes that contribute greatly to the topological potential of the representative point are chosen as the relatively important nodes.

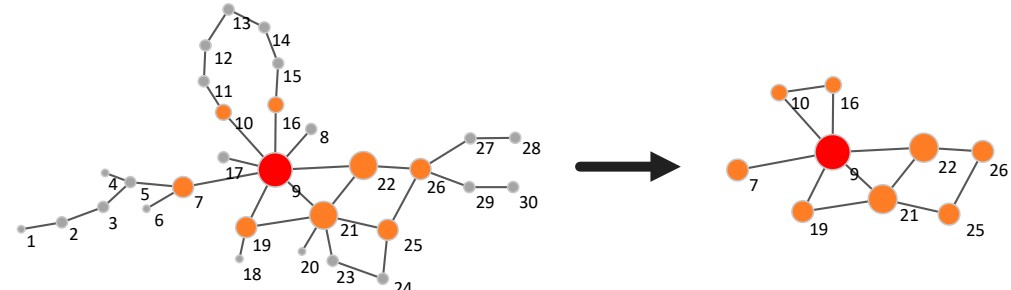

**Figure 2.** Schematic diagram of community node classification and community compression.

### 3.3. Compression Layout Algorithm

When the community structure is compressed, as few nodes as possible are selected to maximize the community structure. The right subfigure in Figure 2 is a schematic diagram for compressing communities on the basis of classification of the community nodes. The community representative point $V_{CA}$ and the relatively important node set $V_{CB}$ are taken as the community structure representative point set $V'_{CA}$. Community compression is mainly to set up alternative nodes for the fringe node $V_{CC}$. It merges fringe nodes and community structure representative points and compresses them into the network, only leaving community structure representative points.

For any node $v_i \in V_{CC}$, $V_N(v_i)$ is used to represent the neighbor nodes set of node $v_i$. When node $v_i$ satisfies Formula (4), node $v_i$ and the community structure representative node $v_j$ are compressed into one node, and $v_j$ is set as an alternative node of $v_i$.

$$v_j = V_N(v_i) \cap V'_{CA}. \tag{4}$$

The breadth-first algorithm is used to find and set the replacement nodes for the nodes in the set $V_{CC}$. The specific method is shown in Algorithm 1, where $If\,Replaced(v_i)$ is used to mark whether node $v_i$ is replaced by a node in $V'_{CA}$, and $replace(v_i)$ is used to represent the replacement point of node $v_i$.

---

**Algorithm 1** Breadth-first traversal instead of node setting algorithm

---

Input: original node $v_i$, neighbor node set $V_N(v_i)$, community structure representative node set $V'_{CA}$.
Output: Replacement node *replace*$(v_i)$

| | |
|---|---|
| 1 | Create a breadth-first access list $L(v_i)$ |
| 2 | WHILE $L(v_i)$ .size>0 DO |
| 3 | $v_j = L(v_i)$ .POP() |
| 4 | IF $(v_j \in V'_{CA})$ |
| 5 | *replace*$(v_i)$ =$v_j$ |
| 6 | *IfReplaced*$(v_i)$ =1 |
| 7 | BREAK WHILE |
| 8 | END IF |
| 9 | ELSE IF |
| 10 | $L(v_i)$.PUSH($V_N(v_j)$) |
| 11 | END ELSE IF |
| 12 | END WHILE |

---

Set up substitution nodes for all nodes in $V_{CC}$, and generate a new community structure representing node set $V'_C$. As shown in Algorithm 1, the alternative node settings for any node $v_i$ in $V_{CC}$ can be divided into two steps:

(1) If the neighbor node $v_j$ of node $v_i$ belongs to the community structure representative node set $V'_{CA}$, then the node $v_j$ is set as the substitution of node $v_i$.

(2) If node $v_j$ is not a representative node of community structure, then all neighbor nodes of node $v_j$ are added to the access list and searched until the replacement node of node $v_i$ is found.

After compressing all fringe nodes in all communities, the new network node firstly sets $V'$ as constructed by merging all the new community structure representative nodes. Then, all the edges in the original network are traversed, the nodes of the edges are replaced by replacement nodes, and the duplicate edges are deleted to obtain the compressed network $G' = (V', E')$. Finally, the classical FR algorithm [28–30] is used to lay out the nodes in the compressed network, resulting in the topology view of the complex network.

## 4. Experiments and Analysis

This paper selects three standard data sets in the field of network community analysis: dolphin, football, and karat. Among them, dolphin is the social network of dolphins, football is the social network of the NCAA football league, and karat is the social network of a karate club. The number of nodes and edges of the corresponding network are shown in Table 1.

The compression layout results of the method of this paper are compared with the network compression layout methods based on node degree value (method 1) and PageRank (method 2). Firstly, the compression effect of different compression methods was quantitatively evaluated through the variations of the number of nodes $|V|$, the number of edges $|E|$, the average clustering coefficient $C$, and the number of communities $N_{comm}$. The average clustering coefficient indicates the degree of aggregation between nodes in the network, and the calculation formula is

$$C = \frac{1}{N} \sum_{i=1}^{N} \frac{2E_i}{k_i(k_i - 1)},$$ (5)

where $E_i$ is the number of the edges that connect to node $i$, and $k_i$ is the degree of node $i$.

Then, by comparing the visual layout effect of the topological structure before and after compression, the advantages of the method in this paper are qualitatively analyzed. The variation of the number of nodes and the number of edges represents the compression scale of the network. The variation of the average clustering coefficient describes the tightness of connection between nodes in

the network. Finally, the variation of the community number reflects whether the compressed network retains the community composition of the original network.

**Table 1.** Comparison of compression results.

| Data | | $|V|$ | $|E|$ | $C$ | $N_{comm}$ |
|---|---|---|---|---|---|
| Dolphin | Before compression | 62 | 159 | 0.303 | 5 |
| | Method 1 | 12 | 29 | 0.716 | 3 |
| | Method 2 | 12 | 29 | 0.716 | 3 |
| | Method of this paper | 13 | 33 | 0.674 | 5 |
| Football | Before compression | 115 | 613 | 0.403 | 10 |
| | Method 1 | 23 | 100 | 0.515 | 9 |
| | Method 2 | 23 | 96 | 0.570 | 8 |
| | Method of this paper | 26 | 97 | 0.534 | 10 |
| Karat | Before compression | 34 | 78 | 0.588 | 4 |
| | Method 1 | 7 | 17 | 0.867 | 3 |
| | Method 2 | 7 | 17 | 0.867 | 3 |
| | Method of this paper | 8 | 18 | 0.733 | 4 |

In the experiment, the selected node compression ratio was 0.2. That is, in methods 1 and 2, the network representative nodes were selected according to 20% of the total number of nodes in the whole network, and the method in this paper selected the representative nodes of the community structure according to 20% of the total number of nodes in each community.

The compression results of the different methods are compared in Table 1. It can be seen that all of the compression methods can reduce the number of nodes and edges to a large extent by merging non-important nodes, thus achieving the purpose of network compression.

In addition, the average clustering coefficient of the compressed network is higher than that before compression, which indicates that the three compression methods above all retain closely connected important nodes, and these nodes and their connecting edges can reflect the skeleton structure of the network. Among them, the average clustering coefficients of the compression networks ranked based on node degree value (method 1) and PageRank (method 2) are relatively close, and both of them are larger than the method in this paper. This is because although the first two methods adopt different node importance evaluation models and the node importance ranking results are not same, for the whole network, important nodes are all distributed in the top ranking region. Therefore, compression results based on the importance of nodes in the global network are the same or basically the same set of representative nodes. In this paper, the importance of nodes in different community structures are sorted respectively. The reserved nodes are important nodes in each community structure and represent the internal structure of each community. Although the average clustering coefficient of the compressed network in this paper is relatively low, the number of communities does not decrease. The community composition of the network is relatively intact, and the overall structure of the network is preserved on the medium scale. The first two methods lost part of the community, resulting in an incomplete overall structure of the network.

Figures 3–5 show the effect pictures of the three methods used to visually compress the layout of different networks. By comparing the layout effects of different methods, it can be seen that the advantages of the method in this paper are mainly reflected in the following two aspects:

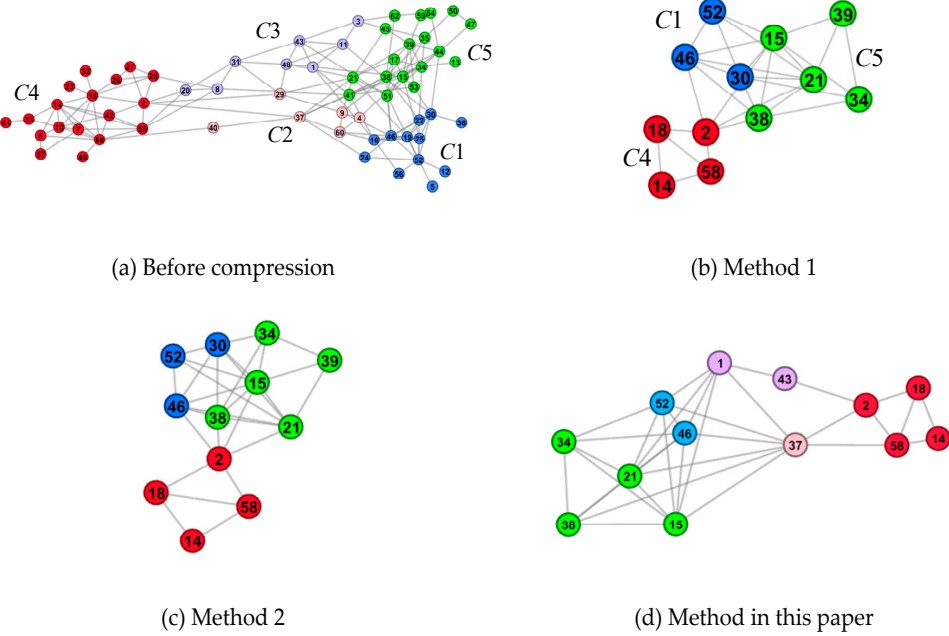

(a) Before compression

(b) Method 1

(c) Method 2

(d) Method in this paper

**Figure 3.** Layout effect of the dolphin network before and after compression.

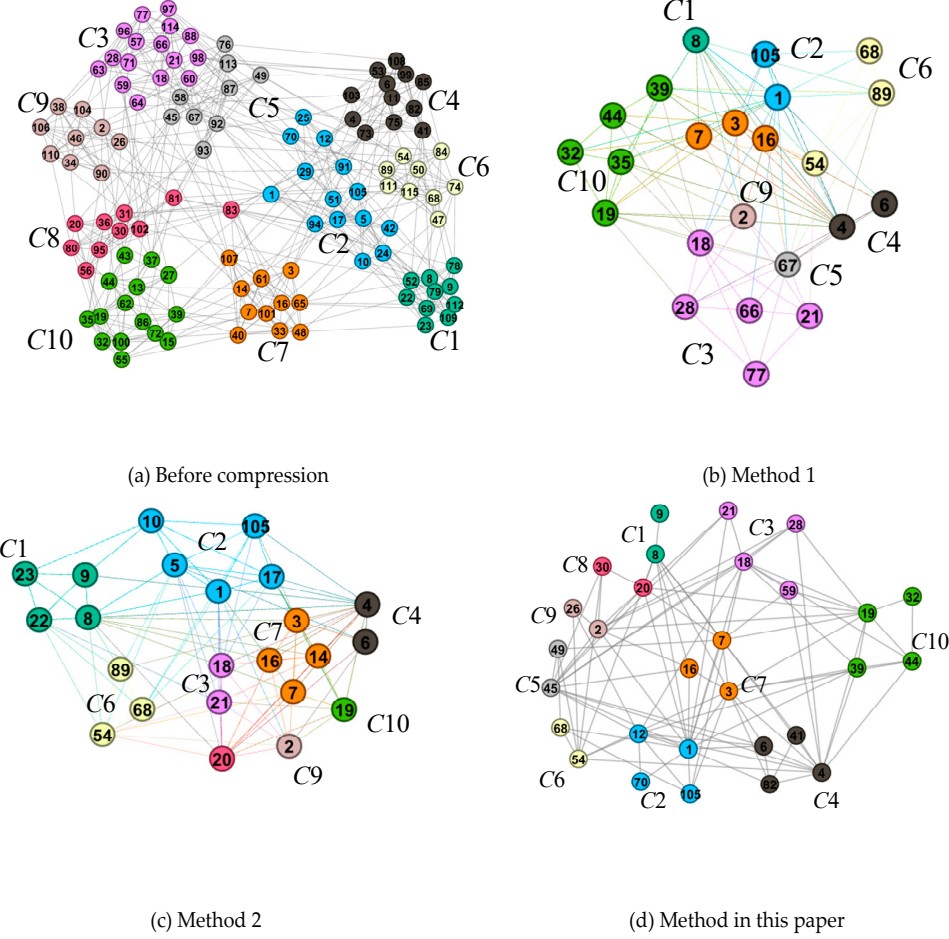

(a) Before compression

(b) Method 1

(c) Method 2

(d) Method in this paper

**Figure 4.** Layout effect of the football network before and after compression.

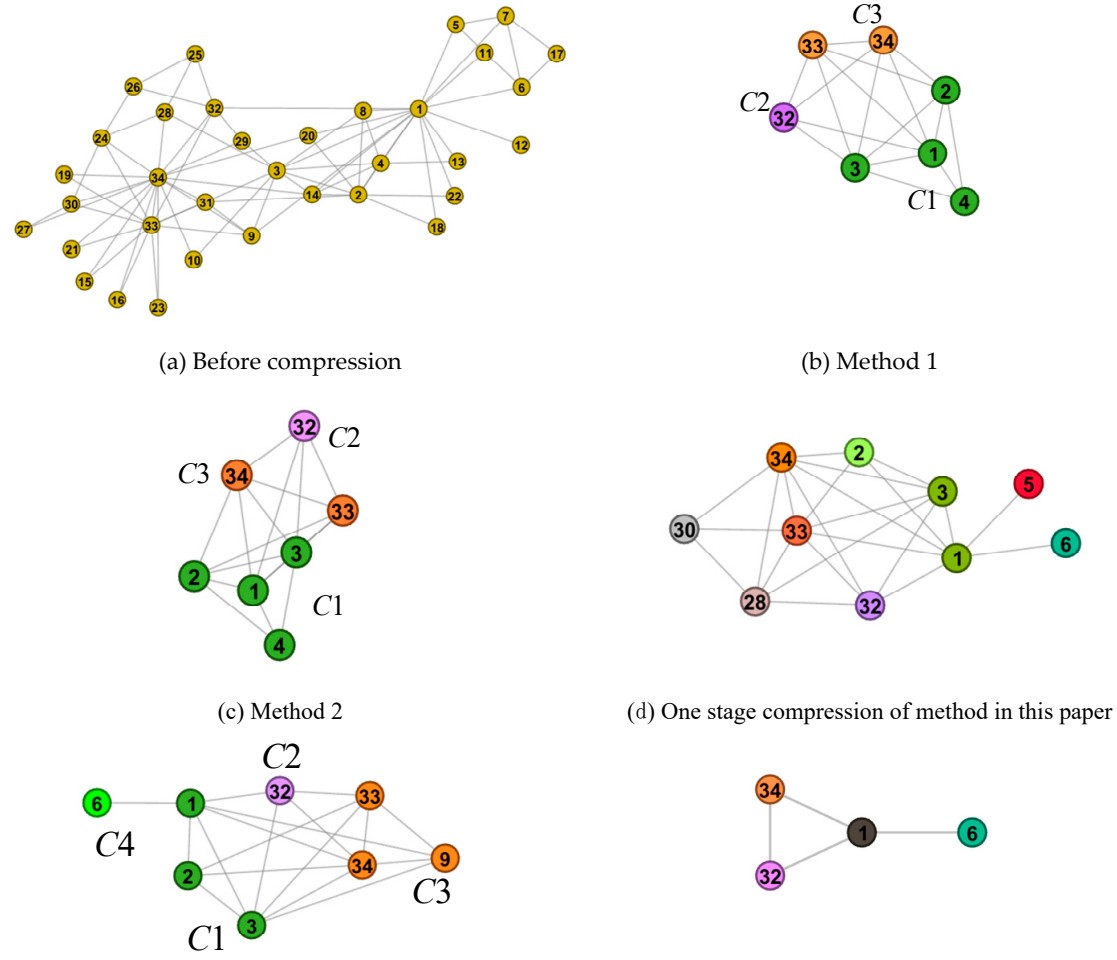

(a) Before compression

(b) Method 1

(c) Method 2

(d) One stage compression of method in this paper

(e) Two stage compression of method in this paper

(f) Three stage compression of method in this paper

**Figure 5.** Layout effect of the karat network before and after multi-granularity compression by the proposed method.

(1) The compression based on the community structure can retain the community composition of the network and highlight the mesoscale structure characteristics.

The sub-graphs (b), (c), and (d) in Figures 3–5 show, respectively, the layout results of the compression network in method 1, method 2, and the method in this paper. Compared with the original network layout results in the sub-graph (a), the number of nodes and the number of edges in the sub-graphs (b), (c), and (d) are compressed to a large extent. The original network in sub-graph (a) is limited by the screen size, with small nodes and dense edges, and the overlap of nodes and edges are serious. In the chaotic overlapping view, it is difficult for users to perceive the overall structure of the network. The three methods mentioned above can reduce the visual clutter caused by overlapping of nodes and edges by reducing the scale of nodes and edges. However, because method 1 (subgraph (b)) and method 2 (subgraph (c)) evaluate the importance of nodes and select network representative nodes based on the global structure, no effective representative nodes are reserved for communities with few nodes or relatively sparse edges. In the process of compression, these communities are compressed and merged, resulting in a decrease in the number of communities, which is unable to accurately display the community composition of the original network. However, the method in this paper (subgraph (d)) is compressed based on the community structure to ensure that each community structure has representative nodes, which avoids the loss of the community in the compression process, and reflects the structural characteristics of the network on the community scale.

In addition, compression based on community structure is more conducive to the perception of the overall structure of the network and the interaction between communities. Taking Figure 3d as an example, it can be found that the original network can be divided into five societies. Among them, $C_1$ and $C_5$ contain a large number of nodes, while the other three have a small number. The interaction between $C_1$ and $C_5$ is mainly realized through $C_2$ and $C_3$. Although there are few nodes in the $C_2$ and $C_3$ communities, they play a "bridge" role in the interaction of nodes in the whole network, which is very important to the topology of the network. However, $C_2$ and $C_3$ communities are not retained in Figure 3b,c, which cannot accurately reflect the "bridge" role of the $C_2$ and $C_3$ communities. $C_4$ directly interacts with $C_1$, which easily causes deviation in users' understanding of the structural features of the network at the mesoscopic scale.

(2) Use of the multi-granularity community structure division algorithm can display the basic structure of the community and realize the multi-granularity compression layout of the network.

In this paper, the fringe nodes are compressed while the necessary community structure representative nodes are preserved. Based on the analysis of the importance of nodes based on topological potential, the reserved nodes make a greater contribution to the community composition, reflecting the basic structure of the community. By retaining these nodes and connections, the basic structure of the community can be clearly displayed. At the same time, by compressing non-important nodes and deleting repeated edges, the number of edges within the community is reduced, which is conducive to the visual perception of the community structure and the discovery of important nodes in the community or network structure.

According to the granularity of community structure division and compression, the original network can be divided into multi-granularity communities. Table 2 shows the results of multi-granularity compression, where the reserved nodes are underlined. Based on the method proposed in this paper, the number of nodes and the number of edges can be reduced by compressing and merging fringe nodes, so as to realize the multi-granularity compression layout. Subgraphs (d), (e), and (f) in Figure 5 show the three-level compression layout for the karat network data.

**Table 2.** Results of multi-granularity compression of the karat network.

| Stage | Communities |
|---|---|
| 1 | $C_1$ = {1, 3, 4, 8, 12, 13, 14, 22}, $C_2$ = {2, 18, 20}, $C_3$ = {5, 7, 11}, $C_4$ = {6, 17}, $C_5$ = {30, 26}, $C_6$ = {32, 25, 29}, $C_7$ = {28, 9, 27, 30, 31}, $C_8$ = {33, 34, 10, 15, 16, 19, 21, 23, 24} |
| 2 | $C_1$ = {1, 2, 3, 4, 8, 12, 13, 14, 18, 20, 22}, $C_2$ = {6, 5, 7, 11, 17}, $C_3$ = {32, 25, 26, 29}, $C_4$ = {9, 33, 34, 10, 15, 16, 19, 21, 23, 24, 27, 28, 30, 31} |
| 3 | $C_1$ = {1, 2, 3, 4, 8, 12, 13, 14, 18, 20, 22}, $C_2$ = {6, 5, 7, 11, 17}, $C_3$ = {32, 25, 26, 29}, $C_4$ = {34, 9, 33, 10, 15, 16, 19, 21, 23, 24, 27, 28, 30, 31} |

In stage 1, nodes in the karat network are divided into eight communities based on the Louvain algorithm, while they are divided into four communities in stage 2. Accordingly, the karat network is compressed into eight communities in sub-graph (d) and four communities in sub-graph (e), by compressing each community. In stage 3, nodes in the karat network are also divided into eight communities. However, the network compression ratio can be controlled according to the node compression ratio, and a community can be compressed into one node at most, as shown in stage 3 of Table 2 and Figure 5f. In Figure 5f, each node represents a community, and "nodes" 1, 32, and 34 can interact directly with each other, while "node" 6 can only interact with other "nodes" through "node" 1. It is possible to observe the mesoscale structural features at different granularities of the network topology from subgraphs (d), (e), and (f) in Figure 5.

## 5. Conclusion

In order to analyze the mesoscale structure characteristics of the network effectively, this paper combined the force-directed layout algorithm with the community structure characteristics of the network, and proposed a method of node layout of a complex network based on community compression. Experimental results and analysis show that this method can effectively compress the number of nodes and edges and reduce visual clutter. At the same time, it can fully reflect the structure and interaction of the network from the mesoscale, so as to analyze the different functions of different communities or nodes in the network. Finally, multi-stage compression layout can be realized based on multi-granularity community structure detection or different node compression ratios. In this paper, the network compression method is combined with the force-directed layout algorithm to visualize the network topology structure from the perspective of composition and structure of communities. In further research, we will consider introducing other node layout algorithms and interactive technologies in order to further optimize the layout results.

**Author Contributions:** C.L. conceived and designed the methods; W.X. guided the students to complete the research; C.L. performed the simulation and experiment tests; X.Z. and Z.L. helped in the simulation and experiment tests; and C.L. wrote the paper.

**Funding:** The authors are grateful for the financial support received from the State 863 Project of China (No. 2014AA7116082) and the Fund Project of Science and Technology on Complex Electronic System Simulation Laboratory (No. 6142010XXXX002).

**Conflicts of Interest:** The authors declare no conflict of interested.

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
