# Peer review of "A Method of Node Layout of a Complex Network Based on Community Compression"

_futureinternet, doi:10.3390/fi11120250_

Round 1

Reviewer 1 Report

The paper proposed a complex network node layout method based on community compression. I think it is interesting. However, there exist many shortcomings. Firstly, the significance and contribution of this paper is not clearly. So, I suggest the authors should highlight these in section Introduction. Secondly, the author use community compression. Why? the reason should be illustrated. Thirdly, in the section related work, I think the author should compare the differences between the literatures and this paper. Finally, more experiments should be added so as to testify the effectiveness of the method proposed in this paper. 

Author Response

Response to Reviewer 1 Comments

Thanks for the reviewers’ time and effort in evaluating my paper. I appreciate your suggestions and valuable comments very much. They are much helpful for improving my paper. I have carefully revised this paper according to your comments. In the following I am going to explain how your comments have been taken into account in the revision. In addition, reviewer’s comments use the black font, and the responses to the comments use the red font. In the revised manuscript, the modified part is also in red font.

Problem1: The significance and contribution of this paper is not clearly. So, I suggest the authors should highlight these in section Introduction.

Answer1:

It has been revised in accordance with the opinions of expert, by pointing out the shortcomings and necessary of the mesoscale display in lines 43~50, lines 98~102, and the significance of this paper in lines 70-76, lines 127-133. Then the relevant references are cited in the corresponding position of the paper.

Problem2: The author use community compression. Why? the reason should be illustrated.

Answer2:

It has been revised in accordance with the expert’s opinion, by pointing out the necessary of the mesoscale visualization in lines 43~46, as well as the analysis of mesoscale structure in complex network. And mesoscale has also been defined in lines 46-48.

What’s more, three types of method for network reduction processing have been overviewed in paragraph 2 on page 2, which can be used to effectively display information and assist users to perceive the mesoscale structure of the network. Compared with filtering and sampling, compression gets the advantages of reducing the scale of the nodes retaining the integrity structure of the network. The composition and other characteristics help users to perceive the overall structure of the network from the mesoscale structure.

Problem3:in the section related work, I think the author should compare the differences between the literatures and this paper.

Answer3:

It has been revised in accordance with the expert’s opinion, by comparing the differences between the node automatic layout methods in paragraph 1 and 2 in “Related Work” section, as well as the differences between the network compression method in paragraph 3 and 4 in “Related Work” section.

In lines 78-94, the basic node layout methods are reviewed. And in lines 95-102 the improved methods are summarized based on the FR model. Finally, the problems of the existing node layout methods have been explained while they are used in community structure displaying and assisting mesoscale analysis. In lines 103-119, this paper summarizes the network compression methods and analyzes their compression effects, and points out the problem that the current method can not effectively support the mesoscale feature analysis.

The network compression method based on topological potential has been reviewed in lines 120-126, and its advantages are explained, which can reflect the positional importance of nodes in the network topology. In 127-133, this paper expounds the idea of the network compression method based on topological potential to the community structure. What’s more, it is combined with the node automatic layout method to form the method idea of this paper, resulting as a visualization method for mesoscale analysis.

Problem4: more experiments should be added so as to testify the effectiveness of the method proposed in this paper.

Answer4:

In the section of “Experiments and analysis”, three datasets are used to testify the effectiveness of the method proposed in this paper. However, the effect of the multi-granularity compression was not display enough in the paper. According to the opinion of experts, a table about the results of multi-granularity compression is added to testify the effectiveness of the Louvain algorithm. What’s more, the explain of the role of the Louvain algorithm has been added in lines 328-343 in the experiment. In general, the advantages of this method proposed are verified from two aspects: (1) compress the number of nodes and edges, resulting as reducing visual clutter in visual view; (2) fully reflect the structure and interaction of the network from the mesoscale, so as to analyze the different functions of different communities or nodes in the network. In addition, combined with the multi-disciplinary community division, this method can better assist the analysis of the mesoscale structural features of the network.

Reviewer 2 Report

This paper proposes a method to analyse complex networks based upon community compression, which can effectively display the mesoscale structure characteristics of the network, making it more convenient for users to analyze the status and function of a single node or a class of nodes in the whole complex network.

The paper introduces the research premise clearly. However, following are some suggested improvements for the paper:

initial proposal and discussion should be more focused around network visualization problem, because that has been used as a basis for the introduction of the paper mesoscale structure, should be explained explicitly for the paper, its relevance for the proposed problem. for the claim on page 3 (line 126-131), proper citation needed line 136 vertex notation is ambiguous, what is difference between vi and i figure 1 is unclear, improve the picture quality. Also, community modularity needs more clarification for how a node is added to a community and when we stop adding nodes in a community. Working of Louvain algorithm, in the example needs clarity. what edges denote, will these properties be preserved if edges are directional?  define average clustering coefficient  FR algorithm needs citation

Author Response

Response to Reviewer 2 Comments

Thanks for the reviewers’ time and effort in evaluating my paper. I appreciate your suggestions and valuable comments very much. They are much helpful for improving my paper. I have carefully revised this paper according to your comments. In the following I am going to explain how your comments have been taken into account in the revision. In addition, reviewer’s comments use the black font, and the responses to the comments use the red font. In the revised manuscript, the modified part is also in red font.

Problem1: Initial proposal and discussion should be more focused around network visualization problem, because that has been used as a basis for the introduction of the paper.

Answer1:

An in-depth review of network visualization is added in the lines 43~50 of “Introduction” section and lines 98~102 of “Related Work” section, pointing out the shortcomings in the mesoscale display, and the significance of this paper is further expounded. The relevant references are as well as cited.

Problem2: Mesoscale structure, should be explained explicitly for the paper, its relevance for the proposed problem.

Answer2:

The research of network synchronization process is actually studying the evolution process of the dynamic behavior of the network from small scale to large scale, which is inseparable from the mesoscale characteristics of complex networks [1]. Therefore, using only one scale to characterize and analyze the dynamic behavior of complex networks is far from adequately describing the complexity of the system. At the small-scale (microscopic, local) level, it is the level of points and edges. The attributes such as degrees, mediations, and clustering coefficients of nodes can be discussed. However, the dynamic behavior of nodes should not be analyzed in isolation and static. The most important feature of the network lies in the mutual coupling and interaction. Therefore, the local micro-scale cannot describe the overall characteristics and regularity of the complex system, and it is not easy to find the physical laws of the complex system as a whole. However, if we only analyze the statistical laws from the large-scale (macro and global) level, that is, from the whole of the network, we can only get the initial state and the final state of the system, and it is easy to cover up the physical process, so it is difficult to find the mechanism of macroscopic law. And the process of evolution. Until now, the existing work of complex networks has been limited to the local scale structure obtained through statistical distribution or the macroscopic scale of the overall parameters of the network, but these two levels can be understood through the intermediate level [1].

Therefore, it is of great theoretical and practical significance to emphasize the mesoscale problem of the research network. The literature [2] pointed out that there is still a large transition space containing various scales between the macro and micro scales. This is called the mesoscale. Mesoscale can be understood as a substructure or subgraph. Substructures have their own topological entities, such as motifs, cliques, cores, loops, or Usually speaking of the community. The literature [3] focuses on the important role of modules or communities in the evolution of network topology and dynamic behavior.

This paper develops a visualization method for mesoscale structures, which is convenient for users to visually observe and analyze the mesoscale structural features of the network, understand the rules and patterns within the network from the mesoscale, and provide support for the research of the synchronization process of the community network.

According to the expert opinion, the definition of mesoscale is given in lines 43~52 of the article, and the relevance of mesoscale in the visualization method of this paper is discussed. The last paragraph of the introduction was modified accordingly.

[1] ALMENDRAL J A, CRIADO R, LEYVA I, et al. Announcement: focus issue on “mesoscales in complex networks” [J]. Chaos, 2010, 20: 010202.

[2] ARENAS A, DíAZ-GUILERA A, PÉREZ-VICENTE C J. Synchronization processes in complex networks[J]. Physica D, 2006, 224: 27-34.

[3] ALMENDRAL J A, CRIADO R, LEYVA I, et al. Introduction to focus issue: Mesoscale in complex networks[J]. Chaos, 2011, 21: 016101.

Problem3: For the claim on page 3 (line 126-131), proper citation needed line 136

Answer3:

It has been revised in accordance with the opinions of expert, respectively in lines 142-147, and 150.

Problem4: Vertex notation is ambiguous, what is difference between vi and i

Answer4:

It has been revised in accordance with the opinions of expert, by correcting i to vi, respectively in lines 145, 173, and 176.

Problem5: Figure 1 is unclear, improve the picture quality.

Answer5:

It has been revised in accordance with the opinions of expert, by redrawing and adding the Figure 1to the paper.

Problem6: Also, community modularity needs more clarification for how a node is added to a community and when we stop adding nodes in a community.

Answer6:

The detailed steps of the Louvain algorithm are described in ref [26-28] and this paper applies it to compression of community structure. To avoid lengthy texts, only the basic ideas of the Louvain algorithm are described. The step of node adding needs to select the target community to be added according to the change of the module degree gain in each cycle, and the specific judgment condition is calculated according to formula (2). The algorithm will end while the module degree gain is not increased. It has been revised in accordance with the opinions of expert, by re-describing the relevant statements in lines 148-151, and lines 165-169.

Problem7: Working of Louvain algorithm, in the example needs clarity.

Answer7:

It has been revised in accordance with the opinions of expert, by adding the results of Louvain algorithm and further explaining the role of the Louvain algorithm in the experiment.

Problem8: What edges denote, will these properties be preserved if edges are directional?  

Answer8:

This problem has been modified, by adding a description of weighted and undirected network in the introduction section.

Complex networks can be divided into directed, undirected, entitled and unlicensed. Different network models are suitable for different specific problem description and analysis needs. The directed edge and the undirected edge respectively correspond to the directed network and the undirected network, and the direction attribute of the connected side of the directed network includes the direction of information or data transmission, for example, the power in the power transmission network is only transmitted from A to B. The node C in the command and control network controls the node D, and the node D can also transmit data to the node C. However, this paper studies the general complex network visualization method and only considers the undirected edge. The different edges of different networks have different physical meanings, but they do not consider their directionality. For example, the edge of the dolphin network indicates the transmission of the dolphin sound signal, and the edge of the football network indicates the match between the American university football team and the edge of the karat network indicates the relationship between the members of the karate club. However the community partitioning method [1, 2, 3] and network visualization method [4, 5, 6] for the directed network are required, while the direction attribute must be considered according to the analysis requirements.

Dolphin social network: an undirected social network of frequent associations between 62 dolphins in a community living off Doubtful Sound, New Zealand. Please cite D. Lusseau, K. Schneider, O. J. Boisseau, P. Haase, E. Slooten, and S. M. Dawson, Behavioral Ecology and Sociobiology 54, 396-405 (2003). Thanks to David Lusseau for permission to post these data on this web site. American College football: network of American football games between Division IA colleges during regular season Fall 2000. Please cite M. Girvan and M. E. J. Newman, Proc. Natl. Acad. Sci. USA 99, 7821-7826 (2002). Zachary's karate club: social network of friendships between 34 members of a karate club at a US university in the 1970s. Please cite W. W. Zachary, An information flow model for conflict and fission in small groups, Journal of Anthropological Research 33, 452-473 (1977).

[1] Xie J, Szymanski B K. Towards Linear Time Overlapping Community Detection in Social Networks[J]. 2012.

[2] Drobyshevskiy M, Korshunov A, Turdakov D. Parallel modularity computation for directed weighted graphs with overlapping communities[J]. Труды ИСП РАН, 2016, 2016:153–170.

[3] YANG Kai, GUO Qiang, LIU Xiao-lu, et al. Detecting Community Structure in Directed Networks Via Multiple Eigenvectors. Journal of University of Electronic Science and Technology of China, 2016, 45(6): 1014-1019, 1032.

[4] Dwyer T, Koren Y, Marriott K. Drawing directed graphs using quadratic programming[J]. IEEE Transactions on Visualization and Computer Graphics, 2006, 12(4):536-548.

[5] Carmel L, Harel D, Koren Y. Drawing Directed Graphs Using One-Dimensional Optimization[C]// Revised Papers from the 10th International Symposium on Graph Drawing. Springer-Verlag, 2002.

[6] Gansner E R, Koutsofios E, North S C, et al. A Technique for Drawing Directed Graphs[J]. IEEE Transactions on Software Engineering, 1993, 19(3):214-230.

Problem9: Define average clustering coefficient

Answer9:

It has been revised in accordance with the opinions of expert, by adding the definition of average clustering coefficient in lines 244-246, as well as the formulas and citations.

Problem10: FR algorithm needs citation

Answer10:

It has been revised in accordance with the opinions of expert, by adding the citations in line 231.

[1] De Leeuw J, Michailidis G. Graph Layout Techniques and Multidimensional Data Analysis[J]. Lecture Notes-Monograph Series, 2000, 35:219-248.

[2] Gephi. http: //www.gephi.org, 2018.

Round 2

Reviewer 1 Report

The authors have addressed all my concerns and I think it may be accepted after English improvement.